# Ancient Origins and Global Diversity of Plague: Genomic Evidence for Deep Eurasian Reservoirs and Recurrent Emergence

**DOI:** 10.3390/pathogens14080797

**Published:** 2025-08-09

**Authors:** Subhajeet Dutta, Aditya Upadhyay, Swamy R. Adapa, Gregory O’Corry-Crowe, Sucheta Tripathy, Rays H. Y. Jiang

**Affiliations:** 1Structural Biology and Bioinformatics Division, CSIR-Indian Institute of Chemical Biology, Kolkata 700032, India; subhajeetdtt@gmail.com (S.D.); adityaupadhyay1997@gmail.com (A.U.); 2Academy of Scientific and Innovative Research (AcSIR), Ghaziabad 201002, India; 3USF Genomics, Global Health Infectious Disease Research Center (GHIDR), Global Health, College of Public Health, University of South Florida, Tampa, FL 33612, USA; 4Harbor Branch Oceanographic Institute, Florida Atlantic University, Fort Pierce, FL 34946, USA

**Keywords:** plague, phylogeny, zoonosis, *Yersina pestis*, pandemic

## Abstract

*Yersinia pestis*, the causative agent of plague, has triggered multiple pandemics throughout human history, yet its long-term evolutionary patterns and reservoir dynamics remain poorly understood. Here, we present a global phylogenomic analysis of ancient and modern *Y. pestis* strains spanning from the Neolithic and Bronze Age to the present day. We show that pandemic-causing lineages did not arise from a single ancestral strain but instead emerged independently along deep branches of the *Y. pestis* phylogeny. Pandemic-associated *Y. pestis* strains were recovered exclusively from human remains and display clear local temporal divergence, indicating evolution driven by human transmission during outbreaks. These findings support the hypothesis that plague emergence is driven by complex, regionally rooted reservoirs, with recurrent spillovers into human populations across millennia. Our work highlights the need to view plague not as a series of isolated outbreaks but as a long-standing zoonotic threat shaped by deep evolutionary history, host ecology, and human societal structures.

## 1. Introduction

*Yersinia pestis*, the bacterial pathogen responsible for plague, has caused some of the most devastating pandemics in human history, including the First Pandemic (541–750 CE) [1,2], the Second Pandemic (14th–18th centuries, including the Black Death) [3,4,5,6], and the Third Pandemic (19th–20th centuries) [4]. Despite over a century of microbiological study and two decades of ancient DNA (aDNA) analysis, major questions remain regarding the evolutionary origin, reservoir ecology, and transmission dynamics that underpin plague’s recurrent emergence across vast geographic and temporal scales.

Recent advances in paleogenomics have enabled the reconstruction of ancient *Y. pestis* genomes from human remains, revealing that the pathogen was already widespread in Eurasia long before the historical pandemics began [7,8,9,10,11]. Notably, strains dating back to the Neolithic and Bronze Age have been recovered from multiple sites [9,10,12], suggesting that *Y. pestis* had a deep-rooted and complex history as both a zoonotic and human-associated pathogen [13,14,15]. However, many of these studies have focused on isolated outbreaks or specific time periods, leaving the global evolutionary landscape of *Y. pestis* incompletely understood.

The First Pandemic represents a critical transition in human history—from sporadic regional outbreaks to the first recorded global pandemic within the known world [16,17,18]. While previously recovered *Y. pestis* genomes from Western Europe mark the geographic and epidemiological fringes of this pandemic, the Jerash site in present-day Jordan [19,20,21,22,23] offers the first direct genetic evidence of plague near the eastern Mediterranean, long recognized as the historical epicenter [24]. The genomes recovered in western Europe came from small numbers of individuals, often single burials within established cemeteries; whereas the Jerash genomes were obtained from multiple individuals interred together in a hastily constructed mass grave. Together, these findings bridge a crucial geographic and temporal gap, providing new insight into the initiation and early spread of the First Pandemic and its movement across interconnected regions of the late antique world.

Here, we present a comprehensive phylogenomic analysis of *Y. pestis* strains ranging from prehistoric times to the modern era, incorporating both previously published genomes and newly reconstructed sequences [25] from a culturally and geopolitically significant Eastern Mediterranean city that hosted a mass grave during the First Pandemic.

By integrating phylogenetic reconstruction, spatiotemporal mapping, and host-range analysis, we address three major questions:How did major *Y. pestis* lineages diverge and persist across millennia?What evolutionary patterns differentiate ancient pandemic strains from modern environmental lineages?What do the geographic origins and host associations of these strains reveal about the nature of plague reservoirs and outbreak dynamics?

Our findings demonstrate that plague pandemics did not stem from a single evolutionary lineage or event but emerged independently from diverse, regionally entrenched reservoirs. The persistence of deeply divergent lineages, combined with the absence of a unified molecular clock, challenges traditional models of linear pathogen evolution and highlights the ecological complexity underlying plague’s long-term history. This work reframes plague not merely as a historical phenomenon but as an ongoing zoonosis shaped by deep time, environmental entrenchment, and human mobility.

## 2. Methods

### 2.1. Genome Selection and Data Acquisition

We have taken ancient as well as modern strains of *Y. pestis* from Genbank for comparative studies. We have taken 258 modern assembled genomes of different biovars such as Orientalis, Medievalis, Antiqua, Pestoides, etc., collected from different geographical locations and from a wide range of hosts (Appendix A). In addition to this, we have taken 68 ancient samples that were available as unassembled raw reads (Appendix A). The ancient samples belonged to different time periods of the global plague pandemic: 19 prehistoric (Late Neolithic Early Bronze Age or LNBA), 16 First Pandemic, 33 Second Pandemic. The metadata associated with the samples, such as the origin of the strain, collection date, etc., were curated from the literature (Appendix A). The latitude and longitude of the geographical locations were calculated using the OpenCage geocoding module v 2.4.0 of Python version 3.11.4 (https://pypi.org/project/opencage/, accessed on 22 June 2024). The *Y. pestis* CO92 genome (Genbank accession number: GCA_000009065.1) was used as the reference strain. The CO92 genome comprises one chromosome and three plasmids (pCD1, pMT1, pPCP1). Two additional genomes of the *Yersinia pseudotuberculosis* (GCA_900637475.1, GCA_000834295.1) were used to serve as outgroups in the tree-building process.

### 2.2. Sequence Alignment and Variant Calling

For the ancient samples, the unassembled reads were used, whereas for the modern samples, the assembled genomes were used. The reads of the ancient samples were pre-processed and subsequently aligned to the reference CO92 *Y. pestis* strain using the nf-core/eager pipeline (v.2.2.1), a Nextflow-based bioinformatics pipeline for ancient genome pre-processing and alignment [26,27]. We used certain filters such as a minimum read length of 35 bp and a minimum 16 bp seed length for bwa aln [28].

The modern assembled *Y. pestis* genomes were aligned with the reference assembly, CO92, using snippy v.4.6.0 (https://github.com/tseemann/snippy, accessed on 24 June 2024), with default parameters and minimum read mapping quality, minimum base quality, and minimum proportion for variant evidence values set to 30, 20, and 0.9, respectively. The snippy output of the ancient samples was evaluated with QualiMap metrics using the bamqc tool of QualiMap v.2.2.2dev [29].

We have used the snippy-core module of the snippy pipeline (v.4.6.0) for generating an alignment file of the core SNPs from the bam files of the modern as well as ancient samples (*n* = 326). The alignment was further refined using a missing data filter to identify and include only those nucleotide positions (sites) where at least 50% of the samples are aligned to the reference. The initial length of the multiple sequence alignment of the core SNPs across all 326 samples was 31,669 bases. After passing it through our missing data filter, the final length of alignment became 31,453 bases, out of which there were 14,003 (44%) singleton sites. This final alignment file was used for phylogenetic tree generation.

### 2.3. SNP-Based Maximum Likelihood (ML) Phylogenetic Analysis

The selection of the best-fit model for the phylogenetic tree was performed using the ModelFinder utility of IQ-TREE multicore v.2.1.2 [30,31]. Here, the best-fit model was “TVM+F+R9” according to Bayesian information criterion (BIC) scores. The ML tree was then constructed using IQ-TREE multicore v.2.1.2 [30,32], where ultrafast bootstrap and SH-aLRT (SH approximate likelihood ratio test) tests were performed for 1000 replicates [33]. The tree file was visualized and annotated using iTOL v.6 [34]. Two genomes of *Yersinia pseudotuberculosis* (GCA_900637475.1, GCA_000834295.1) were used as outgroups in this tree.

### 2.4. Interactive Visualization of Estimated Divergence Tree Using Nextstrain

The dating of our phylogenetic tree was performed using least-square methods to estimate rates with the help of LSD2 v.1.9.8 [35]. Mugration (mutation and migration) analysis was performed on the basis of 4 mugration attributes: “branch major, branch minor, country and province” using mugration argument of treetime v.0.8.1 [36]. Finally, Augur v.24.3.0 [37], Auspice v.2.52.0 [38], and the Nextstrain command-line interface (CLI) v.8.2. [38] were implemented for the interactive visualization of a phylogenetic tree. The tree is hosted at https://nextstrain.org/community/computational-genomics-lab/testHost/oursample, accessed on 25 June 2024.

The entire analysis was carried out using the snakemake pipeline of plague phylogeography (https://github.com/ktmeaton/plague-phylogeography/, accessed on 20 June 2024) [39].

## 3. Results

### 3.1. Spatiotemporal Patterns of Y. pestis Genomes Reveal Deep Eurasian Roots

To reconstruct the early evolution and global spread of *Y. pestis*, we mapped spatial and temporal distributions of ancient and modern genomes across Eurasia, integrating records from prehistory through the First Pandemic (~541–750 CE), alongside modern strain diversity derived from ongoing global surveillance of circulating *Y. pestis* lineages, which are those associated with the Third Pandemic that continues today.

Ancient DNA (aDNA) evidence from the Neolithic and Bronze Age (marked in pink, Figure 1A) spans wide regions from Central Europe to Central Asia, indicating that *Y. pestis* circulated broadly across Eurasia millennia before historically documented pandemics.

Genomes attributed to the First Pandemic (marked in dark blue) are primarily found in Western Europe, representing archaeogenetic confirmation of plague at the far reaches of Roman influence and textual records. In contrast, the Jerash strain—recovered from present-day Jordan—represents the first genetic evidence of plague in the eastern Mediterranean, a region long recognized as the historical epicenter of the Plague of Justinian (Adapa et al., in press) [25]. The recovery of the Jerash plague genome represents the first from an urban center and helps bridge a major geographic and temporal gap in the genetic record of the First Pandemic.

We also incorporated a rare ancient strain from the Tian Shan mountains of Central Asia—recovered from a Hun-associated burial site in present-day Kyrgyzstan—marked as a yellow dot. This strain predates the First Pandemic cluster and likely represents an early diverging lineage circulating on the Central/Western Eurasian steppe, supporting long-range movement and zoonotic transmission prior to pandemic emergence.

To enable dynamic exploration of these genomes, we implemented an interactive NextStrain interface (https://nextstrain.org/community/computational-genomics-lab/testHost/oursample, accessed on 25 June 2024), following methods similar to Eaton et al. Maximum Likelihood (ML) phylogenetic analysis was performed using whole-genome variant data. The optimal evolutionary model was selected based on the Bayesian Information Criterion (BIC), with tree topology supported by 1000 bootstrap replicates. Only bootstrap values of 80 or higher were plotted and used for interpretation.

Phylogenetic reconstruction shows that the First Pandemic strains form a highly supported clade, reinforcing the hypothesis that the pandemic likely originated from the Eurasian steppes and spread rapidly into densely populated urban centers around the Mediterranean (Appendix A). The Jerash genome, located both spatially and temporally near the historical epicenter, represents one of the earliest known large-scale transmission events into a cosmopolitan city. In contrast, the First Pandemic strains recovered from Western Europe likely reflect the later spread and geographic reach of the plague across the fringes of the Roman world.

In contrast, modern *Y. pestis* lineages show strong regional structuring (Figure 1B). Ancestral lineages such as Pestoides (PE) are confined to Central Asia and the Caucasus (purple shading), while more recent biovars—Antiqua (ANT), Medievalis (MED), and Orientalis (ORI)—are found in East Africa, Central Asia, and the Asia-Pacific (yellow shading). These patterns support the view that Central/Western Asia served as both an ancient reservoir and a critical nexus for dispersal during successive plague pandemics.

Together, these data reveal the deep Eurasian roots of *Y. pestis*, highlight complex prehistoric circulation, and show the importance of sampling early and underrepresented regions—such as the Eastern Mediterranean—for understanding the origins and pathways of plague pandemics.

### 3.2. Ancient Evolutionary Patterns of Global Plague Lineages

Our study shows the ancient origins of global plague strains, traced back from the Neolithic and Bronze Age through two pandemics to the present day. Phylogenetic analysis reveals deeply rooted patterns among major modern global plague lineages (Figure 2). The virulent lineages of the First and Second Pandemics do not form an exclusive clade with a single common ancestor but emerged independently on different branches and at different times across a widely divergent *Yersinia* phylogeny. Most modern lineages predate the Black Death, including several, such as PE lineages, which even precede the First Pandemic.

Plague strains exhibit a lack of global relationship between the chronological time of strain presence and pathogen genetic divergence, estimated by the number of changes (mutations) in the genome relative to the root of the phylogenetic tree [38]. This suggests the absence of a single molecular clock (i.e., linear temporal signal) for tracking human infections across the entire span of plague evolution (Figure 3A). Instead, multiple distinct local temporal signals with strain displacement patterns are present (R > ~0.5), as previously reported [39]. During the Neolithic/Bronze Age, First Pandemic, and Second Pandemic, there are linear relationships between strain divergence and time, indicating local temporal displacement correlated with accrued mutations that coincide with intensified human transmissions and historical pandemics. Our analysis also supports the hypothesis of the existence of Neolithic/Bronze Age outbreaks [7], as indicated by identifiable strain displacement patterns—one earlier in the Neolithic/Bronze Age and one later, approximately in the Bronze Age—over several thousand years in prehistory (Figure 3A). This analysis aligns with recent evidence of the Swedish Neolithic plague [11] and suggests such Neolithic/Bronze Age transmissions across large regions of Eurasia.

Major modern plague lineages, despite greater elapsed chronological time, show less divergence compared to strains from prehistoric samples. This suggests a paused or slowed pathogen evolution pattern in modern lineages, possibly due to environmental dormancy in contaminated soil and large zoonotic reservoirs, as plague pathogens can persist in the environment for extended periods of months to a year [40]. This lack of a single global molecular clock suggests challenges in tracking human infections solely based on historical accounts and points to potentially vast zoonotic reservoirs spanning from prehistory to the present day.

The inclusion of SARS-CoV-2 provides a strikingly evolutionary contrast to *Y. pestis*, showing how different pandemic pathogens follow distinct pandemic trajectories. *Y. pestis* persists through ancient, diverse reservoirs, periodically reemerging via multiple independent spillovers across millennia. In contrast, SARS-CoV-2 arose from a single zoonotic event and has evolved primarily through continuous human-to-human transmission, with variant displacement following a linear molecular clock.

These differing dynamics reveal key epidemiological principles. Plague’s episodic resurgence reflects its ecological complexity and adaptation to multi-host systems, whereas SARS-CoV-2’s success hinges on its rapid, uninterrupted spread in humans and adaptation to immune pressures over time. The genetic divergence of SARS-CoV-2 variants correlates closely with time (Figure 3B), unlike *Y. pestis*, where prolonged dormancy and reactivation interrupt clock-like evolution.

### 3.3. Human-Only Host Range in Historical Plague Lineages Suggests Pandemic-Specific Transmission Modes

To further investigate the evolutionary dynamics of *Y. pestis*, we examined the host range associated with each strain across different time periods, from prehistory to the present (Figure 4). Our goal was to determine whether the observed absence of a global molecular clock signal in plague evolution could be explained in part by differences in host association.

This host range analysis revealed a striking temporal trend: strains from prehistoric times, the First Pandemic (541–750 CE), and the Second Pandemic (including the Black Death of the 14th century) were exclusively recovered from human skeletal remains. No corresponding animal or environmental isolates have been identified from these time periods, suggesting that plague outbreaks during these earlier eras may have involved transmission chains largely confined to human populations. This contrasts sharply with modern *Y. pestis* lineages, which have been isolated from a wide range of hosts, including rodents, fleas, and environmental sources such as soil. These lineages—such as the PE, ANT, MED, ORI, and IN clades—appear to persist and circulate within expansive zoonotic reservoirs, contributing to the long-term maintenance and sporadic reemergence of plague in various ecological settings.

The exclusive recovery of historical strains from humans also correlates with the presence of localized temporal strain displacement patterns. During both the First and Second Pandemics, genetically distinct strains replaced one another in relatively short chronological windows. This pattern likely reflects sustained human transmission during pandemic waves, in contrast to the slower and more ecologically buffered transmission dynamics observed in modern enzootic cycles. The appeared absence of environmental or animal reservoirs in ancient pandemics suggests a transmission mode that was more epidemic than endemic, dependent on dense human settlements, trade routes, and sociopolitical factors that facilitated rapid spread and lineage turnover.

Taken together, these findings support a model in which large-scale historical pandemics were driven by human-centric transmission pathways, with limited or no detectable contribution from animal reservoirs. This human-specific transmission mode may help explain the sharper, temporally constrained genetic divergence patterns observed in ancient strains, compared to the slow, reservoir-driven evolution that dominates in the post-medieval and modern eras.

## 4. Discussion

Ancient DNA evidence has pushed the origins of *Y. pestis* much farther back than previously assumed, revealing its presence in Eurasia as early as the Neolithic and Bronze Ages [7,9,10,12]. Our phylogenetic analyses confirm that several ancient strains predate the First and even Second Pandemics, and their genetic divergence is often greater than that of many modern lineages. These early strains appear during critical periods of social and economic transformation, including the rise of agriculture, increasing human mobility, and long-distance trade. These transitions may have facilitated the first sporadic zoonotic transmissions of plague between humans and animals, even before the emergence of fully pandemic-capable forms. The identification of distinct prehistoric clades suggests that plague was not a monolithic, linear evolutionary process but rather a mosaic of localized emergences shaped by early human–animal–environmental interactions. This long, slow, and geographically widespread pre-pandemic evolution set the stage for the pathogen’s explosive emergence during historic pandemics.

Our study emphasizes the critical role of environmental and animal reservoirs in the long-term persistence of *Y. pestis* [13,41]. The patchy and often low-divergence profiles of many modern strains—despite their deep chronological separation from ancient ones—point to extended periods of evolutionary dormancy, likely in soil, fleas, or small mammal hosts. These dormant phases disrupt molecular clock assumptions, creating the appearance of genetic stasis over centuries or millennia. Unlike pathogens that depend on continuous human transmission cycles, *Y. pestis* has evolved to persist in ecologically complex niches, only occasionally spilling over into human populations with devastating effect. This vast ecological flexibility not only explains the unpredictable nature of plague outbreaks but also highlights the importance of surveillance in endemic regions, particularly in Central and Western Asia, where both ancient and modern basal lineages continue to circulate. Understanding this ecological latency is key to deciphering the pathogen’s re-emergence potential and its historical recurrence across widely separated times and places.

While *Y. pestis* may remain genetically inert for centuries within animal or environmental reservoirs, its reentry into dense human populations appears to rapidly accelerate its evolutionary trajectory. We observe distinct episodes of molecular divergence during known pandemic periods, notably the First and Second Pandemics, where human-driven transmission chains likely created strong selective pressures and mutation opportunities. These time-bound bursts of genomic change, contrasted with long periods of stasis, suggest that the molecular clock for plague begins ticking most visibly during large-scale human outbreaks. In this context, humans are not just incidental hosts but active evolutionary engines for plague diversification [16,42]. The concentration of transmission, movement along trade and military routes, and interactions within urban centers collectively provide a biological and social milieu that promotes the emergence of novel variants, such as trade expansion along the Silk Road historically shaping the evolutionary trajectories and spread of *Y. pestis* [3,6]. Thus, the historical record of pandemics is not only a chronicle of suffering but also a record of pathogen evolution catalyzed by human societies themselves.

## 5. Conclusions

Our phylogenetic analysis of *Y. pestis* genomes reveals deep Eurasian origins, broad global diversity, and complex evolutionary dynamics shaped by both human activity and environmental reservoirs. The inclusion of ancient genomes shows that plague pandemics arose through repeated, regionally distinct events—not a single origin. The absence of a global molecular clock and the presence of local temporal signals highlight the non-linear nature of plague evolution. These findings show the need for integrated archaeological, genetic, and ecological approaches to understand the emergence and re-emergence of ancient pathogens.

## 6. Limitations

The limitations of our study include uneven geographic and temporal sampling of *Yersinia pestis* genomes, particularly from underrepresented regions such as Central and Western Asia and North Africa. Many ancient genomes are derived from human remains, while corresponding environmental and animal reservoirs remain poorly sampled, limiting our understanding of the full host range and transmission ecology.

## Figures and Tables

**Figure 1 pathogens-14-00797-f001:**
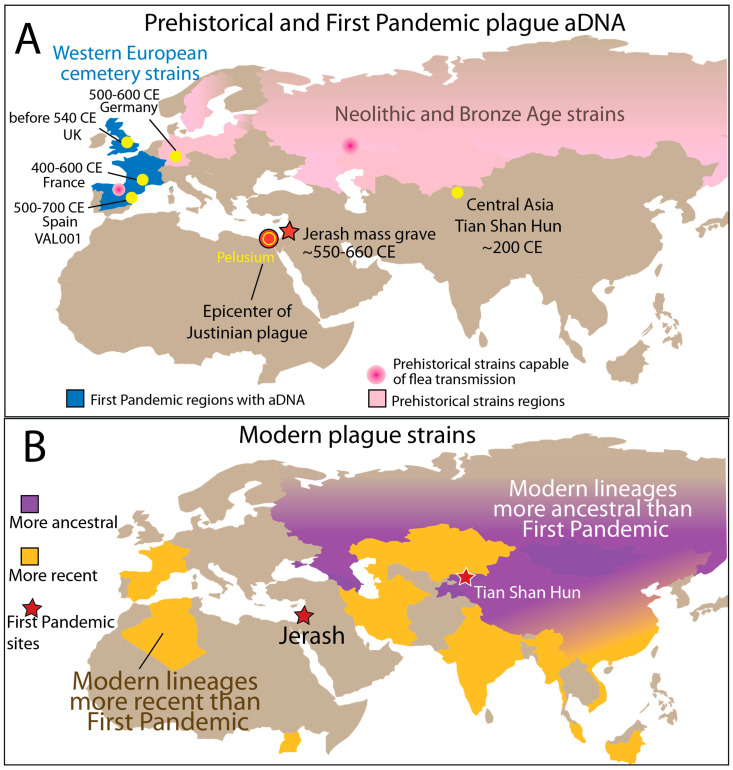
Geographic and temporal distribution of *Yersinia pestis* genomes across Eurasia from prehistory to historical pandemics. (**A**) Locations of ancient *Y. pestis* DNA identified to date are shown, with sites from the Neolithic and Bronze Age marked in pink and those from the First Pandemic period (~541–750 CE) in dark blue. The Jerash site (star) represents the earliest archaeogenetic evidence of plague near the historically documented epicenter of the First Pandemic in the Eastern Mediterranean. Yellow dots within dark blue regions indicate published First Pandemic genomes from Western European cemeteries. A single genome from the Tian Shan region (yellow dot) represents a potential precursor strain circulating in Central or Western Asia prior to the Justinianic Plague. (**B**) Geographic distribution of modern plague lineages suggests a Central/West Asian origin. Modern lineages have also been identified in other regions, including North America and Madagascar (not shown). Areas with more ancestral modern lineages (e.g., Pestoides [PE]) are shaded in purple, while those dominated by more recent lineages (e.g., Antiqua [ANT], Medievalis [MED], and Orientalis [ORI]) are shaded in yellow, reflecting historical patterns of dispersal and diversification.

**Figure 2 pathogens-14-00797-f002:**
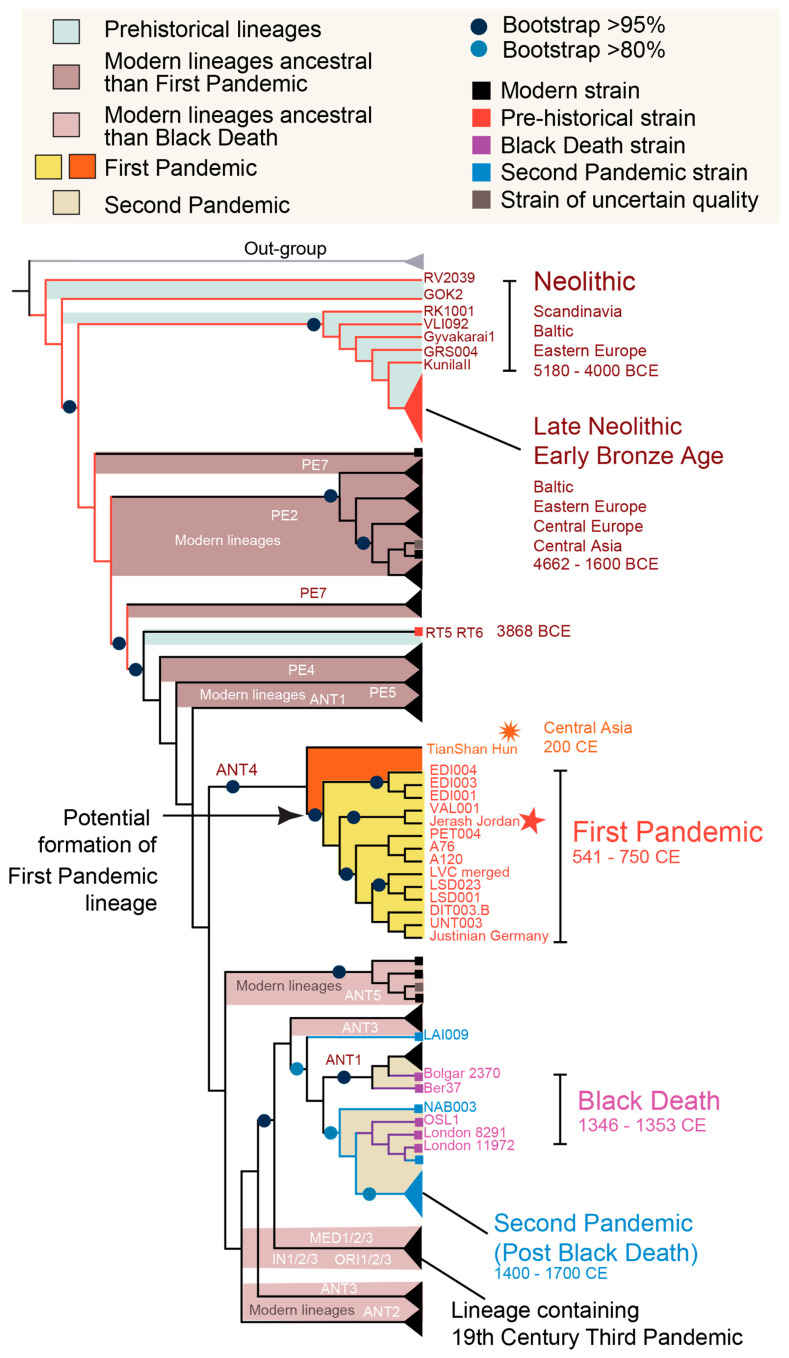
Global phylogenetic analysis of *Y. pestis* from prehistory to the present. Maximum Likelihood phylogenetic reconstruction of *Y. pestis* genomes, with tree topology supported by 1000 bootstrap replicates. The phylogeny reveals deep-rooted patterns, highlighting the ancient origins of modern global plague lineages and indicating that many prevalent lineages began diverging early in plague evolutionary history. A Central Asian strain is clustered with First Pandemic genomes recovered from the eastern Mediterranean to Western Europe, forming a highly supported clade. Notably, most major modern lineages predate the Black Death, and several—such as the Pestoides (PE) lineage—even precede the First Pandemic. All modern lineages refer to genomes from current *Y. pestis* collections.

**Figure 3 pathogens-14-00797-f003:**
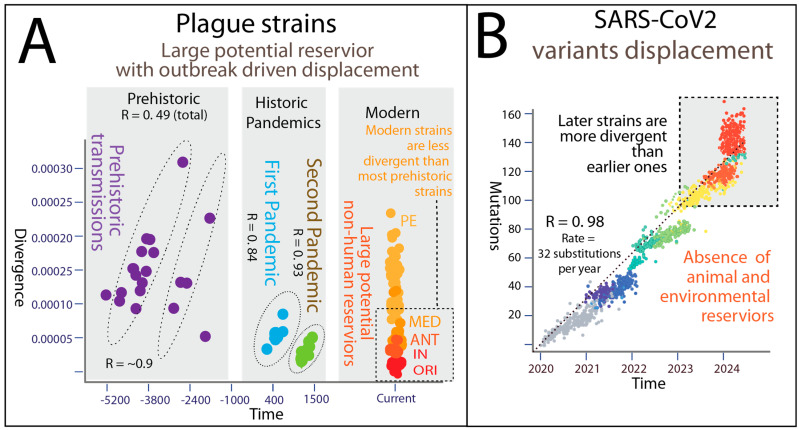
Contrasting evolutionary dynamics of *Yersinia pestis* and SARS-CoV-2. Genetic divergence or mutation counts are plotted on the *y*-axis against chronological time. A linear increase in divergence over time indicates the presence of an active, time-dependent molecular clock. In contrast, flat or shallow divergence despite the passage of time reflects a slower or interrupted molecular clock, consistent with periods of evolutionary dormancy or reduced mutation rates. (**A**) Plague strains: Plague strains exhibit a lack of global relationship between the chronological time of strain presence and pathogen genetic divergence, indicating the absence of a single global molecular clock for tracking human infections throughout plague evolution. Instead, multiple distinct local temporal signals with strain displacement patterns are observed. Linear relationships between strain divergence and time are evident during Prehistory, the First Pandemic, and the Second Pandemic, reflecting transmissions associated with intensified human interactions and historical pandemics. The prehistorical strain divergences show correlations in two timeframes: one earlier in the Neolithic/Bronze Age and another later in the Bronze Age. Major modern plague lineages (boxed in dotted lines), despite the greater chronological time elapsed, exhibit less genetic divergence compared to most prehistoric strains. This suggests a paused or slowed molecular evolution in modern lineages, potentially due to environmental dormancy and extensive zoonotic reservoirs. (**B**) SARS-CoV-2 strains: SARS-CoV-2 strains demonstrate a strong correlation between mutational load and time, indicating lineage displacement over time. Colour indicates chronological differences in strain isolation. The molecular clock of SARS-CoV-2 remains active in human infections without significant animal reservoirs or environmental dormancy. Newer strains post-2023 exhibit greater genetic divergence compared to earlier COVID-19 pandemic strains.

**Figure 4 pathogens-14-00797-f004:**
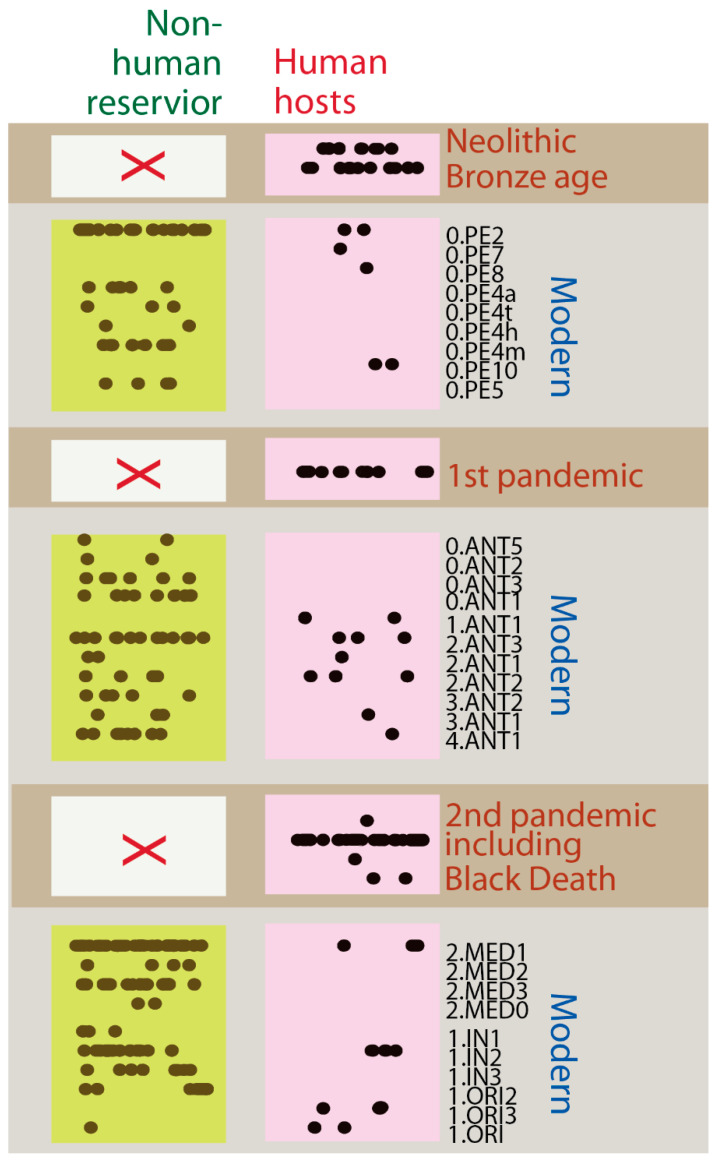
Host range analysis of *Y. pestis* strains reveals human-specific associations in historical pandemics. Temporal host range analysis shows that modern plague lineages (e.g., PE, ANT, MED, ORI, and IN) are found in both human and non-human hosts, including animals and environmental reservoirs. In contrast, strains from prehistory, the First Pandemic, and the Second Pandemic (including the Black Death) have only been recovered from human remains. The absence of non-human isolates during these earlier periods is indicated by red X marks, suggesting that historical outbreaks were driven by human-to-human transmission, with no direct evidence of zoonotic or environmental reservoirs during those pandemics.

## Data Availability

The data presented in this study are available in NCBI as listed in Appendix A. These data were derived from the following resources available in the public domain, NextStrain: https://nextstrain.org/community/computational-genomics-lab/testHost/oursample, accessed on 25 June 2024.

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
