# Peer review of "Ancient Origins and Global Diversity of Plague: Genomic Evidence for Deep Eurasian Reservoirs and Recurrent Emergence"

_pathogens, 2025, doi:10.3390/pathogens14080797_

Round 1

Reviewer 1 Report

Comments and Suggestions for Authors

This study provides a seminal contribution to plague genomics by integrating spatiotemporal, phylogenomic, and host-dynamic evidence across millennia, fundamentally reshaping our understanding of plague as a recurrent zoonosis rooted in deep evolutionary reservoirs.

There are some suggestions regarding revisions to the plague genomics paper, incorporating the requested modifications:

  1. This paper analyzed the global phylogeny of ancient and modern plague strains from the Neolithic and Bronze Age to the present. According to my understanding, the modern plague should include the data of the second and third world plague pandemics, but there is no relevant statement about strains of the third world plague pandemic in the whole article, including the key results in Figure 1 and Figure 2.
  2. The published data of Yersinia pestis strains in the world has reached thousands, while this study included only less than 100 This paper aims to conduct in-depth analysis based on the global Yersinia pestis genome diversity, whether these strains are sufficiently representative, and whether this is the reason why the third world plague pandemic strains were not included for analysis.

Reviewer 2 Report

Comments and Suggestions for Authors

The manuscript by Subhajeet Dutta and co-authors titled Ancient Origins and Global Diversity of Plague: Genomic Evidence for Deep Eurasian Reservoirs and Recurrent Emergence is devoted to a global phylogenomic analysis of ancient and modern genomes of the plague bacterium using state-of-the-art bioinformatics methods. The manuscript is important and interesting but requires significant revision/re-submission before acception.
It should be noted that the authors clearly rushed the submission, as their results rely on their own unpublished article (reference 26), which they have presumably submitted to some journal. The authors should wait for that article to be published before submitting the present manuscript. Moreover, the journal's Instructions for Authors explicitly prohibit citations to unpublished works.
Additionally, the authors merged sections that should not be combined - namely, Methods and Results, while unnecessarily relegating Materials to the supplementary information. The journal imposes no length restrictions, as it is an online publication. Therefore, the Materials and Methods described in the supplementary section should be formatted as a proper section, and the entire manuscript should adhere to the IMRAD structure.
The comparison with COVID-19 is artificially forced, seemingly to inflate the manuscript’s length.

Further comments and suggestions are provided below:
Line 34 and throughout: Italicize Yersinia pestis and Y. pestis.
Citations should be in square brackets and not italicized.
Line 113: Broken link: "The community repository, dataset, or narrative "nextstrain.org/community/computational-genomics-113" doesn't exist."
Figure 4 should appear after its first mention in the text, not after Figure 3.
Lines 208, 216, 319: After the first mention, genus names should be abbreviated to the first letter followed by a period (e.g., Y. pestis).
Missing sections: Funding, Author Contributions, and Supplementary Materials list.
Supplementary Materials: Italicize Yersinia pestis and Y. pestis. Citations should be in square brackets.

Reviewer 3 Report

Comments and Suggestions for Authors

The manuscript by Subhajeet Dutta and colleagues investigates genomic diversity and ancient origin of Yersinia pestis, causative agent of plague, focusing particularly on deep Eurasian reservoirs and repeated emergence events throughout human history. This research topic is relevant and timely given ongoing global interest in the evolution and epidemiology of infectious diseases, especially pathogens with historical significance like Y. pestis. The study utilizes detailed phylogenomic analyses integrating ancient and modern genomes, presenting interesting evidence that plague pandemics arose independently from diverse, regionally rooted reservoirs rather than single lineage. The manuscript is really interesting, but could be better organised and structured. It is supported by illustrations and figures, which contribute understanding, but their quality could be also improved.

Overall, despite minor points, the manuscript provides valuable insights into plague evolution. 

Specific Comments:

Lines 51-58. Additional details on Jerash historical importance related specifically to plague transmission would strengthen the argument.

Section 2. This section should be either be divided into two sections “Materials and Nethods” and “Results” or include two distinct subsections. All parameters of software used should be clearly presented.

Figure 2: Please indicate the bootstrap values better, using numbers and omitting the values under 50%.

Lines 169-186: The comparative analysis of Y. pestis and SARS-CoV-2 is intriguing, however, its purpose in this manuscript feels slightly disconnected. Either integrate this comparison more directly into plague evolutionary discussion or shorten this section to maintain focus.

Lines 300-307: The discussion rightly emphasizes the role of human societies in pathogen evolution, but a clearer distinction or explicit examples of societal actions that significantly shaped pathogen diversification would enrich this argument. However, I think it is a very nice ending for the manuscript.

Figure 3 - please improve the quality of the Figure and explain everything written in the picture in caption.

Round 2

Reviewer 2 Report

Comments and Suggestions for Authors

Citations should be in square brackets and not italicized.

Materials and Methods should be placed before the Results.

Add title to the table S1: Supplemental Materials: Supplement table S1

Format the References accordingly to the Instructions for Authors.

Change status of the paper S. R. Adapa et al., Genetic Evidence of Yersinia pestis from the First Pandemic. (submitted) [ref. 26] to accepted and add the journal, year, number, pages.

Reviewer 3 Report

Comments and Suggestions for Authors

The revised manuscript has been clearly improved, presenting a more complete, scientifically sound, and well-supported account. The authors have addressed the previous feedback carefully and introduced several important modifications. 
